# The Effect of Radiotherapy on Cell Survival and Inflammatory Cytokine and Chemokine Secretion in a Co-Culture Model of Head and Neck Squamous Cell Carcinoma and Normal Cells

**DOI:** 10.3390/biomedicines11061773

**Published:** 2023-06-20

**Authors:** Sybilla Matuszczak, Krzysztof Szczepanik, Aleksandra Grządziel, Alina Drzyzga, Tomasz Cichoń, Justyna Czapla, Ewelina Pilny, Ryszard Smolarczyk

**Affiliations:** 1Center for Translational Research and Molecular Biology of Cancer, Maria Sklodowska-Curie National Research Institute of Oncology, Gliwice Branch, 44-102 Gliwice, Poland; 2Radiotherapy Department, Maria Sklodowska-Curie National Research Institute of Oncology, Gliwice Branch, 44-102 Gliwice, Poland; 3Radiotherapy Planning Department, Maria Sklodowska-Curie National Research Institute of Oncology, Gliwice Branch, 44-102 Gliwice, Poland

**Keywords:** radiotherapy, HNSCC, fibroblasts, endothelial cells, apoptosis, IL-6, tumor microenvironment

## Abstract

Radiotherapy (RT) is one of the main treatments for head and neck squamous cell carcinomas (HNSCCs). Unfortunately, radioresistance is observed in many cases of HNSCCs. The effectiveness of RT depends on both the direct effect inducing cell death and the indirect effect of changing the tumor microenvironment (TME). Knowledge of interactions between TME components after RT may help to design a new combined treatment with RT. In the study, we investigated the effect of RT on cell survival and cell secretion in a co-culture model of HNSCCs in vitro. We examined changes in cell proliferation, colony formation, cell cycle phases, type of cell death, cell migration and secretion after irradiation. The obtained results suggest that the presence of fibroblasts and endothelial cells in co-culture with HNSCCs inhibits the function of cell cycle checkpoints G_1_/S and G_2_/M and allows cells to enter the next phase of the cell cycle. We showed an anti-apoptotic effect in co-culture of HNSCCs with fibroblasts or endothelial cells in relation to the execution phase of apoptosis, although we initially observed increased activation of the early phase of apoptosis in the co-cultures after irradiation. We hypothesize that the anti-apoptotic effect depends on increased secretion of IL-6 and MCP-1.

## 1. Introduction

Radiotherapy (RT) is an effective and widely used cancer treatment. It is one of the main treatment methods for patients with head and neck squamous cell carcinomas (HNSCCs). Early-stage disease is typically treated with RT or surgery. Locoregionally advanced disease is usually treated with a combination approach, including surgery followed by RT and/or chemotherapy [1,2]. Despite the recent progress in RT, patient outcomes remain poor, and radioresistance is still observed in many cases. Many studies indicate that the effectiveness of RT depends on both the direct effect inducing cell death and the indirect effect of changing the tumor microenvironment (TME). The TME consists of a heterogeneous population of tumor cells, the extracellular matrix (ECM) and wide variety of other non-tumor cells that have lost the ability to maintain tissue homeostasis and architecture specific to healthy tissue. The population of non-tumor cells contains fibroblasts, cancer-associated fibroblasts (CAFs), mesenchymal stromal cells, adipocytes, immune cells, blood and lymph vessels, including endothelial cells and pericytes [3,4,5]. Autocrine and paracrine communication between cells in a tumor constantly changes the TME profile and adapts it to new conditions. TME cells are responsible for the escape or activation of immune surveillance, formation of blood and lymphatic vessels, oxygenation, cell survival and mobility. Recently, it has been better understood that TME components play a key role in determining the success or failure of therapy, because some TME cells may exhibit both pro- and anticancer properties, depending on their activation status [3,4,5,6,7].

The largest population of non-tumor cells in the TME is CAFs. For example, late-stage HNSCCs often consists of up to 80% CAFs. The high presence of CAFs in HNSCCs is strongly correlated with cancer progression, relapse, metastasis, treatment resistance and poor prognosis for patients [2,8]. They play a significant role in tumor biology. CAFs have pro-tumor properties by stimulating the growth and survival of cancer cells, inducing the migration of other cells to the TME, especially immune cells and cells involved in angiogenesis, modulating the innate and acquired immune response, and regulating metabolic activity in the TME and cancer cell mobility [4,5,8,9,10,11]. CAFs are a heterogeneous cell population, and some CAFs subtypes show anti-tumor functions by limiting tumor growth and migration [5,8,12]. Both CAFs and normal tissue fibroblasts are radioresistant cells capable of surviving high doses of radiation. After high-dose (>10 Gy) radiation, they develop an irreversible senescence phenotype, whereas low doses of radiation induce reversible DNA damage without growth arrest in vitro. Senescent CAFs change the secretory profile, which is considered to be responsible for tumor radioresistance [8,10,12].

Other cells within the TME that play an important role in tumor progression and treatment response are endothelial cells. They form vessels that control tumor oxygenation and stimulate cells within the TME, intensify inflammatory reactions, regulate the activity of immune cells, and allow cancer cells to avoid apoptotic death [5,7]. Endothelial cells are characterized by a rapid proliferation rate, which contributes to their radiosensitivity. Single high-dose (>10 Gy) radiation induces endothelial cell death, leading to increased vessel permeability, detachment from the underlying basement membrane, reduced vascular density and, in consequence, the development of thrombosis, hypoxic and immunosuppressive TME. This indirectly causes cell death within the TME, resulting in tumor volume reduction. A low radiation dose (<10 Gy) promotes vascular relaxation and increases tumor oxygenation, which improve tumor radiosensitivity. This is also associated with a stimulation of angiogenesis and neovascularization [10,12].

Prior research indicates that RT can act as a double-edged sword on cancer. Thus, to successfully eradicate cancers, it is important to know the reciprocal interactions between the TME components, especially their response to irradiation. Detailed knowledge of these interactions will help design combinations of RT with other targeted therapy to increase radiosensitivity and to avoid undesirable pro-cancer effects. Therefore, in the current study, we investigated the effect of RT on cell survival and inflammatory cytokine and chemokine secretion in the co-culture model of HNSCCs. We observed anti-apoptotic effects in co-cultures of fibroblasts or endothelial cells with HNSCCs in relation to the execution phase of apoptosis, which is probably dependent on increased secretion of IL-6 and MCP-1.

## 2. Materials and Methods

### 2.1. Cell Culture and Irradiation

Studies were carried out on HNSCCs cell lines: FaDu (human squamous cell carcinoma, ATCC, Manassas, WV, USA), A253 (human submaxillary salivary gland carcinoma, ATCC) and normal cell lines: HUVECs (Human Umbilical Vein Endothelial Cells, PromoCell GmbH, Heidelberg, Germany) and Wi-38 (caucasian fibroblast-like fetal lung cell, ECACC). Cells were maintained in: FaDu in DMEM high glucose (Biowest, Nuaillé, France) supplemented with 10% heat-inactivated FBS (EURx, Gdansk, Poland) and 1% penicillin–streptomycin (Biowest, Nuaillé, France), A253 in McCoy’s 5A Medium Modified (Gibco BRL, Paisley, UK) supplemented with 10% heat-inactivated FBS and 1% penicillin–streptomycin, HUVECs in Endothelial Cell Growth Medium with SupplementMix (PromoCell GmbH, Heidelberg, Germany) supplemented with 20% FBS and Wi-38 in Minimum Essential Medium Eagle (Merck, Darmstadt, Germany) supplemented with 10% heat-inactivated FBS, MEM Non-Essential Amino Acids Solution (Gibco BRL, Paisley, UK) and 1% penicillin–streptomycin. Cell cultures were kept in standard conditions: 37 °C, 5% CO_2_, and 95% humidity in incubator. Cells were passaged with 0.25% trypsin-EDTA (Biowest, Nuaillé, France) every 3–4 days. For irradiation, cells were placed in 24- or 48-well plates. Studies were carried out on cancer cell monocultures and co-cultures of cancer cells with normal cells. Co-cultures of cancer cells with Wi-38 and the compared monocultures were maintained in DMEM high glucose supplemented with 10% heat-inactivated FBS and antibiotics. Co-cultures of cancer cells with HUVEC and the compared monocultures were maintained in Endothelial Cell Growth Medium with SupplementMix supplemented with 20% FBS. Cells were irradiated with single dose: 2 or 10 Gy in a phantom, at a depth of 10 cm. Photon beam of 6 MV with dose rate of 400 MU/min from TrueBeam accelerator (Varian Medical Systems, Palo Alto, CA, USA) was used. Irradiation was performed at the Department of Radiotherapy, National Research Institute of Oncology, Gliwice Branch. The dose distribution planning was prepared by the medical physicist, using Eclipse treatment planning system (Varian Medical Systems, Palo Alto, CA, USA) in the Department of Planning Radiotherapy.

### 2.2. Double-Stranded DNA Breaks after Radiation

The day before irradiation, cells were placed in 24-well plates at density 6 × 10^5^ cancer cells per well (monocultures) or 3 × 10^5^ cancer cells and 3 × 10^5^ normal cells per well (co-cultures). Immediately after, irradiation cells were detached by 0.25% trypsin and washed with PBS^-^. Next, cells were fixed and permeabilized by 70% ice-cold ethanol and stored at −20 °C. Next day, cells were washed and incubated with anti-γH2AX PE antibody (BioLegend, San Diego, CA, USA) for 30 min in dark. Mean fluorescence γH2A.X PE intensity was determined using BD FACSCanto™ II flow cytometer (BD, Franklin Lakes, NJ, USA).

### 2.3. Cell Proliferation and Radiosensitivity

The day before irradiation, cells were placed in 48-well plates at density 5 × 10^4^ cancer cells per well (monocultures) or 5 × 10^4^ cancer cells and 5 × 10^4^ normal cells per well (co-cultures). Three days later, cell proliferation and radiosensitivity were analyzed using the MTS assay (Promega, Madison, WI, USA) according to the manufacturer’s instruction. Reagent solution was added to each well and incubated for 2 h at 37 °C. The absorbance was measured at 490 nm on Microplate Reader (TECAN, Männedorf, Switzerland).

### 2.4. Colony Formation Assay

Immediately after, irradiation cells were detached by 0.25% trypsin and placed in 6-well plates at cell density 4 × 10^2^ cells per well. Cells were kept in standard conditions: 37 °C, 5% CO_2_, and 95% humidity in incubator for 7 days. Then cells were fixed by ice-cold methanol (10 min) and dried overnight. Next day, cells were stained for 30 min with 0.25% gentian violet solution (PPF HASCO-LEK S.A., Wroclaw, Poland). Colonies containing 50 cells or more were counted using brightfield microscopy Nikon Eclipse 80i (Nikon Instruments Inc., Tokyo, Japan). The plating efficiency was calculated according to the formula: (number of colonies counted/number of cells plated) × 100%.

### 2.5. Cell Cycle Analysis

The day before irradiation, cells were placed in 24-well plates at density 4 × 10^5^ cancer cells per well (monocultures) or 2 × 10^5^ cancer cells and 2 × 10^5^ normal cells per well (co-cultures). Three days after irradiation, cells were detached by 0.25% trypsin and washed with PBS^-^. Cells were fixed and permeabilized with 70% ice-cold ethanol and stored at −20 °C until further analysis. Next, cells were washed with PBS and incubated with anti-CD90 FITC (co-cultures of FaDu with Wi-38 cells (BioLegend, San Diego, CA, USA), anti-CD10 APC (co-cultures of A253 with Wi-38, BioLegend) and anti-CD31 FITC (co-cultures with HUVECs, BD Biosciences, San Jose, CA, USA) for 30 min in dark. Then, cells were incubated in PBS^-^ containing RNase A (1 mg/mL, (Qiagen, Venlo, Netherlands)) and propidium iodide (PI, 1 mg/mL, (Cayman Chemical, MI, USA)) for 1 h at 37 °C. The DNA content of cancer cells was analyzed using BD FACSCanto™ II flow cytometer.

### 2.6. Apoptosis and Cell Death Determination

The day before irradiation, cells were placed in 24-well plates at density 4 × 10^5^ cancer cells per well (monocultures) or 2 × 10^5^ cancer cells plus 2 × 10^5^ normal cells per well (co-cultures). Three days after, irradiation cells were detached by 0.25% trypsin, and cell apoptosis was detected using PE Annexin V Apoptosis Detection Kit I (BD Biosciences, Franklin Lakes, NJ, USA) according to the manufacturer’s instructions. In brief, cells were resuspended in binding buffer and stained with Annexin V and 7-AAD at room temperature for 15 min in dark. Finally, cells were analyzed using FACSCanto™ II flow cytometer. Additionally, one day later, cells were stained using anti-Cleaved Caspase-3 antibody (Cell Signaling Technology, Denvers, MA, USA) followed by secondary antibody Texas red-conjugated anti-rabbit (Vector Laboratories, Burlingame, CA, USA) and DAPI dye (Merck) administration. Microscopic observations were performed using an LSM710 confocal microscope (Carl Zeiss Microscopy GmbH, Oberkochen, Germany). The results were defined as the area occupied by stained cells calculated using ImageJ 1.48v software (National Institutes of Health, Bethesda, MD, USA).

### 2.7. Migration Assay

Immediately after, irradiation cells were detached by 0.25% trypsin and placed in 96-well plates at density 5 × 10^4^ cells per well. Cells were cultured up to 90–100% confluency (about 24 h) and scratched with sterile plastic pipette tip across the monolayer. Cell debris was removed by washing with medium. Images were taken after wounding every 10 min for 24 h using Zeiss Cell Observer SD microscope (Carl Zeiss Microscopy GmbH, Jena, Germany). The migration distance was measured using ZEN 2012 software version 3.2 (Carl Zeiss Microscopy GmbH, Jena, Germany) and calculated according to the formula: wound width at 0 h—wound width after 6 or 12 h (µm).

### 2.8. Inflammatory Cytokine and Chemokine Secretion

The day before irradiation, cells were placed in 24-well plates at density 4 × 10^5^ cancer cells per well (monocultures) or 2 × 10^5^ cancer cells and 2 × 10^5^ normal cells per well (co-cultures). Immediately after, irradiation medium was changed and left for 72 h. The type and quantity of cytokines and chemokines secreted by cells into conditioned medium were assessed using LEGENDplex™ Human Inflammation Panel 1 (analyzed cytokines and chemokines: IL-1β, IFN-α2, IFN-γ, TNF-α, MCP-1, IL-6, IL-8, IL-10, IL-12p70, IL-17A, IL-18, IL-23, IL-33; BioLegend) according to the manufacturer’s instruction. Samples were analyzed on BD FACSCanto II flow cytometer. Raw data were analyzed using LEGENDplex^TM^ Data Analysis Software Version 8.0 (BioLegend, San Diego, CA, USA). Furthermore, IL-6 and MCP-1 were examined using Human IL-6 ELISA Kit II (BD Biosciences, Franklin Lakes, NJ, USA) and ELISA MAX^TM^ Deluxe Set Human MCP-1/CCL2 (BioLegend, San Diego, CA, USA) according to the manufacturer’s instruction. Absorbance was measured on Microplate Reader.

### 2.9. Statistics Analysis

The results were statistically analyzed with appropriate tests. The normality of the distribution was verified with the Shapiro–Wilk test. The homogeneity of variance was checked using the Brown–Forsythe and/or Levene’s tests. The Student’s *t*-test or the Mann–Whitney *U* test was used to compare monocultures with co-cultures depending on the variable distribution and variance homogeneity. Statistical analysis was performed using Statistica software version 12 (TIBCO Software Inc., StatSoft Polska, Kraków, Poland) and *p*-values < 0.05 were considered statistically significant.

## 3. Results

### 3.1. Radiosensitivity and Cell Proliferation Activity in Cell Co-Cultures after Irradiation

In order to evaluate the number of DNA double-strand breaks (DSBs), γ-H2AX staining was performed. The γ-H2AX staining is the most precise method for detecting DNA damage after irradiation and evaluates DSBs formation in a 1:1 ratio [13]. Immediately after irradiation, the number of DSBs was determined using anti-γH2AX antibody and flow cytometry analysis. DSBs increased in all groups after 2 and 10 Gy irradiation (Figure 1a). The highest number of breaks in all cultures was observed after 10 Gy irradiation. We observed about 2–2.5-times more DSBs in FaDu monocultures compared to A253 monocultures after 2 or 10 Gy irradiation. Furthermore, the presence of DSBs after 2 and 10 Gy irradiation decreased in co-culture of FaDu with Wi38 compared to the FaDu monocultures but without statistical significance.

Next, we examined the radiosensitivity of monocultures and co-cultures after irradiation in short-term (MTS assay) and long-term (colony formation assay) tests. Three days after irradiation, cell proliferation was analyzed using MTS assay. Cell proliferation increased in co-culture of FaDu with HUVECs compared to the FaDu monoculture after 10 Gy irradiation but without statistical significance. No changes in cell proliferation were observed in co-cultures of both cancer cell lines with Wi-38 fibroblasts compared to cancer cell monocultures with or without irradiation (Figure 1b). In general, higher radiosensitivity of cells cultured in endothelial cell-specific medium (Endothelial Cell Growth Medium with SupplementMix and with 20% FBS) than the same cells cultured in fibroblast-specific medium (DMEM high glucose with 10% FBS) was observed. After irradiation, the proliferation ability of HUVECs decreased by 20–35%, respectively, for a dose of 2 or 10 Gy. No effect of radiation on the proliferation of Wi-38 cells was observed. Seven days after irradiation, the ability of colony formation decreased in co-cultures of both cancer cell lines with Wi-38 and A253 with HUVECs compared to the cancer cell monocultures after 2 Gy irradiation and in non-irradiated groups (Figure 1c). After 10 Gy irradiation, the plating efficiency in all co-cultures and monocultures did not exceed 2.5%, and there were no statistically significant differences between them. Additionally, the increased ability of A253 to form colonies in the non-irradiated groups and after 2 Gy irradiation was observed when they grew in an endothelial cell-specific medium compared to the same cells grown in a fibroblast-specific medium.

DSBs formation after irradiation induces cell cycle arrest in the G_1_/S or G_2_/M cell cycle checkpoints. It serves to protect genomic integrity and prevents cells with damaged DNA from entering S phase or M phase until DSBs are repaired [1]. Three days after irradiation, the percentage of cancer cells in particular phases of the cell cycle was analyzed. In order to distinguish cancer cells from fibroblasts or endothelial cells in co-cultures, supplementary surface antigen staining, CD90 (specific for Wi-38 cells compared to FaDu cells), CD10 (specific for Wi-38 cells compared to A253 cells) and CD31 (specific for HUVEC cells compared to cancer cells), was performed. In co-cultures of FaDu with Wi-38 and A253 with HUVECs, the percentage of cancer cells in the S phase of the cell cycle were increased and the percentage of cells in the G_2_/M phase was decreased compared to cancer cell monocultures after 2 or 10 Gy irradiation (Figure 1e). Moreover, in co-culture of A253 with HUVEC, the percentage of cancer cells in the SubG_1_ phase of the cell cycle was increased. The same trends were observed in co-cultures of A253 with Wi-38 and FaDu with HUVECs after 2 or 10 Gy irradiation compared to cancer cell monocultures, but the results were not statistically significant.

### 3.2. Cell Death Determination in Cell Co-Cultures after Irradiation

DNA damage after radiation causes cell cycle arrest followed by senescence or mitotic catastrophe or cell death through apoptosis, necrosis or autophagy-dependent cell death. Three days after irradiation, cells were analyzed using the PE Annexin V Apoptosis Detection Kit I. This assay was used to show the early and late phases of apoptosis or necrosis. After 10 Gy irradiation, in co-cultures of cancer cells with HUVECs and A253 with Wi-38, the percentage of cells in the early phase of apoptosis increased compared to the percentage of cancer cells in the monocultures (Figure 2a). This trend was also observed in co-cultures of A253 with Wi-38 compared to cancer cell monocultures and in all co-cultures after 2 Gy irradiation, but these results were not statistically significant. Moreover, in the late phase of apoptosis or necrosis, no differences in the percentage of cells were observed between all co-cultures and the corresponding monocultures. The percentage of cells in the early phase of apoptosis was 2–2.8-times higher in co-cultures with HUVECs compared to co-cultures with Wi-38 after 2 Gy irradiation and 1.5–3.5-times higher after 10 Gy irradiation. The percentage of cells in the early phase of apoptosis was 6–8-times higher in HUVECs monocultures compared to Wi-38 monocultures with or without irradiation. Additionally, four days after, irradiation cells in all groups were stained with an antibody against cleaved caspase-3 (activated caspase-3). Caspase-3 is one of the effector caspases and plays a key role in the execution phase of apoptosis. The area of caspase-3-positive cells decreased in co-cultures of cancer cells with Wi-38 and A253 with HUVECs compared to cancer cell monocultures after 2 or 10 Gy irradiation (Figure 2d). A similar trend was observed in all co-cultures compared to cancer cell monocultures, but the results were not statistically significant for all.

### 3.3. The Migration Ability in Cell Co-Cultures after Irradiation

One of the hallmarks of cancer cells that promote disease progression is the ability of cell migration. One day after irradiation, cell migration was analyzed using the scratch-wound assay. Observations were carried out for 24 h using Zeiss Cell Observer SD microscope. In co-cultures of cancer cells with HUVECs, cell migration was slower than in cancer cell monocultures for the first 12 h after 10 Gy irradiation (Figure 3b). Migration slowdown was also observed for the first 6 h, but it was statistically significant only for co-cultures of A253 with HUVECs (Figure 3a). Likewise, after 2 Gy irradiation in co-cultures of A253 with HUVECs, cell migration was slower than in monocultures for the 6 and 12 h of the test, but these results were not statistically significant. No changes in cell migration rate were observed in co-cultures of cancer cells with Wi-38 fibroblasts compared to cancer cell monocultures with or without irradiation.

### 3.4. Inflammatory Cytokine and Chemokine Secretion in Cell Co-Cultures after Irradiation

Tumors are often regarded as “wounds that never heal” because chronic inflammation within the TME is observed. Various cells present in the TME are indirectly responsible for this effect by secreting numerous cytokines and chemokines. This is important because chronic inflammation and tumor progression are closely related [4]. Inflammation cytokines and chemokines secreted by tested cells into the conditioned medium during 72 h after irradiation were assessed using the LEGENDplex™ Human Inflammation Panel 1 via flow cytometric analysis. Cells in all co-cultures secreted increased the amount of IFN-α2, IL-6 and MCP-1 into the medium compared to cancer cell monocultures with or without irradiation (Figure 4a). The exception is cells from co-cultures of A253 with Wi-38 compared to A253 monocultures, where no statistically significant increase in MCP-1 secretion was observed. Also, in co-cultures of A253 with HUVECs and FaDu with Wi-38, the increase in IL-6 secretion was not statistically significant compared to monocultures after irradiation. The amount of secreted IL-6 and MCP-1 increased above the upper limit measured by LEGENDplex^TM^ Human Inflammation Panel 1; therefore, additional determinations of this cytokine and chemokine were performed using ELISA kits. Detailed analysis using ELISA Kits showed increased secretion of IL-6 and MCP-1 into the medium in all co-cultures compared to cancer cell monocultures with or without irradiation (Figure 4c). In both tests, the exception is co-culture of A253 with HUVECs, where no statistically significant increase in IL-6 was detected compared to the A253 monoculture after 10 Gy irradiation (Figure 4a,c). Simultaneously, in co-cultures of A253 with HUVECs, the increase in IL-12p70 and IL-23 cytokine secretion was observed compared to cancer cell monocultures with or without irradiation (Figure 4a). The same statistically significant results were observed for co-cultures of FaDu with HUVECs but only after 10 Gy irradiation. Furthermore, high IL-8 secretion (above the upper limit of the assay) was observed in all co-cultures and monocultures, with the exception of A253 monocultures, with or without irradiation.

## 4. Discussion

RT has been used for over a century to treat cancer patients. The standard radiation dose used for treatment is approximately 2 Gy per fraction five times a week for a few weeks [14]. The concept of dose fractionation is designed based on the rationale that the rapidly proliferating cancer cells are more sensitive to radiation than normal cells. Cancer cells also have defective DNA repair mechanisms and aberrant checkpoint function during the cell cycle, leading to their death or senescence after one dose of radiation, while normal healthy cells have a chance to repair their DNA [7]. Unfortunately, due to the radioresistance of tumors and other application limitations, such as tumor location, further research on the use of radiotherapy is necessary.

Radiation can act directly by inducing DNA damage or indirectly via highly reactive oxygen species (ROS) generated by the ionization of water inside the cells [6,11]. The most cytotoxic lesions generated after irradiation are DSBs. A standard fractionating dose of 2 Gy is capable of inducing approximately 3000 DNA breaks per cell, but only 40 DSBs are needed to induce cell death [1]. In the present study, we observed the increase in DSBs in all irradiated groups (Figure 1a). The highest number of breaks in all cultures was after 10 Gy irradiation. The obtained results showed no differences in the number of DSBs either in cells from monocultures or from co-cultures with fibroblasts and endothelial cells. The results confirm the correctness of the irradiation model used in all experiments in this study.

In the present study, we examined two HNSCCs cell lines with different origins to confirm whether the observed results are common to HNSCCs in general. According to place of origin, the FaDu cell line belongs to the group of hypopharyngeal carcinomas and A253 cell line to the group of oral cavity carcinomas [15]. We observed a greater radiosensitivity of FaDu cells compared to A253 cells (Figure 1a). HNSCCs belong to cancers containing a high number of CAFs, which is associated with a high-grade malignancy and a poor prognosis for patients [10,16]. The high abundance of CAFs, leukocytes and endothelial cells in rectal tumors was associated with poor prognosis and predicted resistance to RT [17]. Therefore, we decided to choose fibroblasts (Wi-38) and endothelial cells (HUVECs) for our co-culture model in this study.

In the first part of the study, we assessed the radiosensitivity and radioresistance of selected cells using a co-culture model and cancer cell monocultures, analyzing cell proliferation, colony formation, phases of cell cycle and cell death. Cell proliferation analysis showed HUVECs radiosensitivity unlike Wi-38 cells (Figure 1b). Furthermore, we did not observe changes in cell proliferation in all co-cultures compared to cancer cell monocultures with or without irradiation. In a long-term assay, the ability of colony formation decreased in co-cultures compared to cancer cell monocultures after 2 Gy irradiation and in non-irradiated groups (Figure 1c). We did not distinguish cancer cells from fibroblasts or endothelial cells; therefore, we suppose that the colonies formed could be a mixture of cells in the co-cultures. Similarly, Steer et al. observed colonies composed of a mixture of cancer cells and fibroblasts or colonies of cancer cells alone, but they did not find colonies composed only of fibroblasts [16]. In our cell cycle analyses, we observed an increase in the percentage of cancer cells in the SubG_1_ and S phases and a decrease in the G_2_/M phase in co-cultures compared to cancer cells of monocultures (Figure 1d). These observations were not statistically significant for all cell co-cultures compared to monocultures, but we observed the same trend for all of them. The obtained results suggest that the presence of fibroblasts and endothelial cells inhibits the function of cell cycle checkpoints G_1_/S and G_2_/M and allows cells to enter the next phase of the cell cycle (S phase or M phase, respectively).

DNA damage after irradiation most often leads to cell death (by apoptosis, necrosis, autophagy-dependent cell death) or non-lethal processes (mitotic catastrophe and cellular senescence). Although cells have different sensitivity to radiation, low doses of radiation are believed to induce apoptosis, while high doses are more specific for inducing cell necrosis [18]. After 10 Gy irradiation in co-cultures with HUVECs, we observed an increase of the percentage of cells in the early phase of apoptosis compared to the percentage of cancer cells in the monocultures (Figure 2a). This trend was observed in co-cultures with Wi-38 compared to monocultures and after 2 Gy irradiation for all co-cultures, but these observations were not statistically significant. The obtained results were unexpected. We observed greater activation of the early phase of apoptosis in cells from co-cultures compared to cancer cells from monocultures three days after irradiation, but one day later, it seemed that the executive phase of apoptosis was inhibited. Four days after irradiation, we observed a decrease in caspase-3-positive cells in all co-cultures compared to cancer cell monocultures after 2 or 10 Gy irradiation (Figure 2d). These observations were not statistically significant in all groups. We also observed that HUVEC cells undergo apoptosis more easily compared to Wi-38 cells, again indicating a higher radiosensitivity of HUVEC cells. Adjemian et al. observed, in a panel of distinct mouse cancer cell lines (colon, cervical, lung adenocarcinoma, fibrosarcoma), a dose-dependent increase in cells in the early phase of apoptosis and caspase activity after radiation [19]. Gehrke et al. showed that co-culture of FaDu with pre-irradiated fibroblasts significantly decreased FaDu viability and increased cancer cell apoptosis compared with co-culture with non-irradiated fibroblasts and monocultures. Pre-irradiated fibroblasts were isolated from skin radiated with 60–70 Gy a few months before the experiments [20].

Our results showed slower cell migration in co-cultures of HNSCCs with HUVECs compared to cancer cell migration in monocultures for the first 12 h after irradiation (Figure 3b). Similarly, Kim et al. observed that IL-6 secreted by endothelial cells induced epithelial–mesenchymal transition (EMT) and migratory phenotype of HNSCC cells [21]. However, it should be noted that our migration assay was carried out in co-cultures of HNSCC cells in direct contact with endothelial cells, whereas, in the abovementioned studies, migration assays were performed using a conditioned medium from endothelial cells or with a mechanical barrier between endothelial and cancer cells. This difference could significantly affect the obtained results. In the co-cultures of HNSCC cells with Wi-38 fibroblasts, faster migration was only observed after 6 h in co-culture of A253 cells. In other groups, no changes was observed in the cell migration rate compared to cancer cell monocultures (Figure 3a,b). Suzuki et al. showed faster migration of HNSCC cells (SAS and FaDu cell lines) in co-cultures with fibroblasts, especially after their irradiation. They also confirmed that the migration of HNSCC cells was induced by fibroblast-secreted IL-6 [22]. However, unlike our methodology, the Suzuki group performed a migration assay with a mechanical barrier between HNSCCs and fibroblasts.

Normal fibroblasts can exert suppressive functions against cancer cells through direct cell–cell contact, paracrine signaling and ECM stability. Loss of the suppressive function by fibroblasts is the first step in tumor progression and results from activation of fibroblasts to CAFs by tumor cells [23]. There is no single precise marker to identify CAFs populations; therefore, the differences between CAFs and normal fibroblasts in the TME should be considered functional [24]. CAFs display many pro-tumor properties by secreted cytokines, chemokines, growth factors, exosomes, ECM proteins and ECM-degrading enzymes (e.g., IL-6, TNF-α, CXCL12, TGF-β, VEGF, IGF, collagen, matrix metalloproteinases) or, less frequently, show anti-tumor functions depending on their activation status [4,5,8]. The CAFs phenotype is changing during tumor progression and after anti-cancer therapies. Until now, CAFs plasticity and function have been the subject of research. Like fibroblasts, endothelial cells also secrete specific growth factors (e.g., PDGF, VEGF and FGF), which act as survival factors and a wide range of pro-inflammatory cytokines and chemokines (e.g., IL-1β, IL-6, TNF-α, CXCL1 and CCL2) [5]. Changes in the expression of cytokines, chemokines and growth factors within the TME may affect the progression of many cancers, including HNSCCs. Recent studies have investigated the role of cytokines in regulating tumor growth, as prognostic markers and as therapeutic targets in HNSCCs [25]. In our study, we observed an increased amount of IL-6, MCP-1 and IFN-α2 secretion in all co-cultures compared to cancer cell monocultures with or without irradiation (Figure 4a,c). The exception is co-culture of A253 with HUVEC after 10 Gy irradiation.

IL-6 is a cytokine that regulates various homeostatic and pathological processes. It was identified as a cytokine abundantly present in the TME of various tumor types, including HNSCCs, pancreatic cancer, non-small-cell lung cancer, breast cancer, ovarian cancer and melanoma [21,26,27]. In cancer, IL-6 stimulates proliferation, survival, angiogenesis, TME immunosuppression, radioresistance, invasiveness and metastasis. Notably, the IL-6/STAT3 signaling pathway activates the transition from the G1 to S phase of the cell cycle, leading to cell proliferation, and induces the expression of the anti-apoptosis proteins Bcl-2, Bcl-xl, Mcl-1 and survivin, which inhibits apoptosis and promotes proliferation and angiogenesis [26,28,29,30]. Yadav et al. showed that endothelial cells co-cultured with HNSCC cells (CAL27) secrete greater amounts of IL-6, which induces EMT of tumor cells [31]. These observations are in line with our results of IL-6 secretion in co-culture of HNSCC cells with endothelial cells with or without irradiation (Figure 4a,c). In our study, we observed a significant increase in IL-6 secretion in HNSCCs with fibroblast co-cultures compared to cancer cell monocultures with or without irradiation (Figure 4a). The highest amount of IL-6 was observed in co-cultures of FaDu with Wi-38 cells (Figure 4c). In addition, we observed significantly higher IL-8 secretion (above the upper limit of the assay) in all co-cultures of HNSCCs with Wi-38 fibroblasts or HUVEC endothelial cells and in FaDu monocultures, with the exception of the A253 monocultures, with or without irradiation (Figure 4a). Chen et al. also observed that external or internal stimulation of FaDu cells with IL-6 causes radiation resistance [32]. Recent data indicate that IL-6 secreted by cancer-associated fibroblasts in a mouse model of human breast cancer MDA-MB-231 induces tumor radioresistance, confirming our observations in vitro [33]. The authors also showed that the inhibition of IL-6 receptor by specific neutralizing monoclonal antibody abrogated CAFs-induced growth and radioresistance.

MCP-1 is a chemokine secreted in high amounts by cells in all co-culture models compared to cancer cell monocultures with or without irradiation (Figure 4a,c). The highest amount of MCP-1 was observed in co-cultures of HNSCCs with HUVECs. Ebrahimian et al. also observed an increase in IL-6 and MCP-1 secretion by HUVECs after 2 Gy irradiation [34]. The main function of MCP-1 is the recruitment of blood monocytes to inflammatory sites or tumors, which contributes to the increase in the amount of tumor-associated macrophages within the TME. RT promoted the secretion of MCP-1 by cells within the TME. After RT, MCP-1 together with IL-6 mediated radiation-induced macrophage infiltration [35,36]. Furthermore, in the HNSCCs microenvironment, a high amount of MCP-1 induced the recruitment of activated regulatory T lymphocytes, which suppress tumor-associated antigen effector T-cell immune responses and inhibit the effectiveness of RT in HNSCCs patients [37,38].

Moreover, in co-cultures of A253 with an HUVECs increase in IL-12p70 and IL-23, cytokine secretion compared to cancer cell monocultures with or without irradiation was observed (Figure 4a). The same statistically significant results were observed for co-cultures of FaDu with HUVECs after 10 Gy irradiation. IL-12 and IL-23 are closely related cytokines that antagonistically regulate inflammatory responses in the TME. IL-12 promotes anti-tumor immunity due to its role attracting and activating natural killer cells, T helper 1 (Th1) and cytotoxic T lymphocytes (CD8^+^), whereas IL-23 promotes the activity of regulatory T lymphocytes, expansion of T-helper 17 cells (Th17), angiogenesis and reduces cytotoxic CD8^+^ T lymphocytes in the TME. In addition, IL-23 stimulates neutrophil and macrophage infiltration in the TME [39,40,41]. Kortylewski et al. observed the activation of IL-23 secretion and inhibition of IL-12 secretion in both tumor cells and infiltrating myeloid cells within the TME at the same time [41]. Lee et al. suggested that RT inhibits IL-12 secretion through the irradiation-induced IL-6 secretion in tumors [42]. These observations are inconsistent with our results, indicating increased secretion of all these cytokines simultaneously.

It should also be noted that the significant reservoir for cytokines, chemokines and growth factors in tumors is the ECM. Additionally, the ECM can regulate tumor response to radiation by influencing oxygen availability. Oxygen is critical for RT response because molecular oxygen can stabilize ROS, enabling them to cause DNA damage for a long period of time. Hypoxic cells are generally 2.5−3-times less radiosensitive than normoxic cells [4,6,7]. Therefore, it seems reasonable to test the complex interactions of tumor cells with different tumor microenvironment cells using more sophisticated research models. The limitation of our work is that experiments were performed using 2D models of cell lines, which provide basal information about tumor biology. More complex 3D co-culture models could better mimic the tumor architecture. Thus, 3D systems involving multicellular tumor spheroids would be the next research model to study the complex interactions between tumor microenvironment cells after radiation exposure.

## 5. Conclusions

In the present study, co-culture models with fibroblasts or endothelial cells demonstrate pro-tumor and anti-apoptotic effects on HNSCC cells in vitro. Interactions in co-culture between fibroblasts or endothelial cells with HNSCC cells change the secretome and the ability to survive after irradiation. We showed anti-apoptotic effects in the co-culture of fibroblasts or endothelial cells with HNSCCs in relation to the execution phase of apoptosis, although we initially observed increased activation of the early phase of apoptosis in the co-cultures after irradiation. We hypothesize that the anti-apoptotic effect depends on the increased secretion of IL-6 and MCP-1, but further studies are needed to confirm this. The obtained results are a good starting point for further studies on the cytoprotective role of IL-6 and MCP-1 against HNSCCs, first in 2D and 3D in vitro models, based on an in vivo model, designing an appropriate anti-cancer therapy combined with radiotherapy.

## Figures and Tables

**Figure 1 biomedicines-11-01773-f001:**
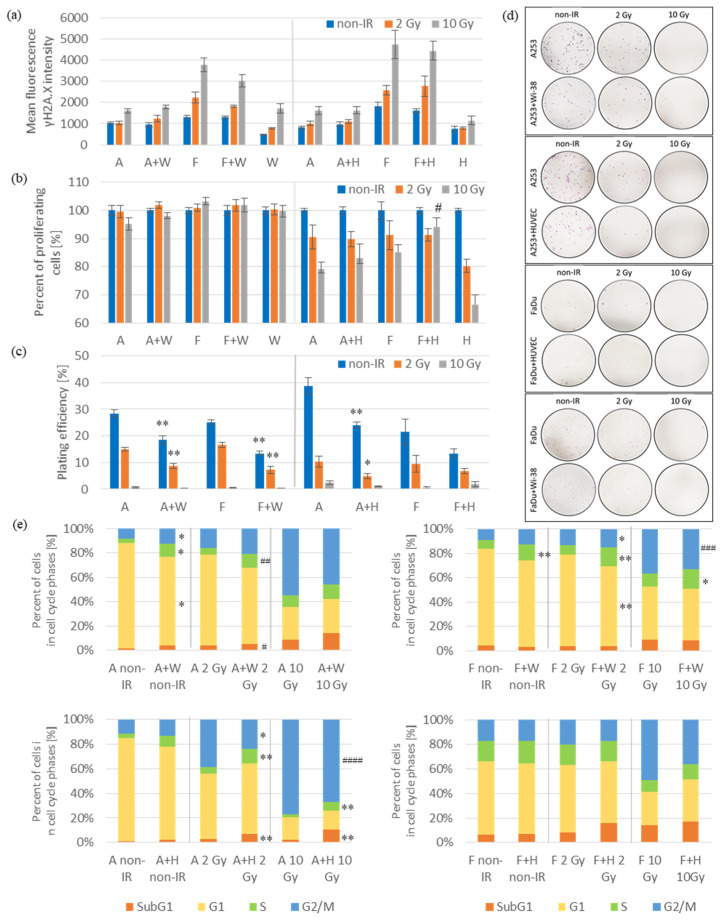
Radiosensitivity and cell proliferation activity in cell co-cultures compared to cancer cell monocultures after irradiation. (**a**) Immediately after irradiation, the number of DSBs was determined using anti-γH2AX antibody staining. Mean fluorescence intensity was analyzed using flow cytometer. Results are shown as mean ± SEM (*n* = 3–4). (**b**) Three days after irradiation, cell proliferation was analyzed using the MTS assay. Results are shown as the percentage of proliferating cells and are expressed as mean ± SEM (*n* = 6–9; ^#^
*p* = 0.055). (**c**) Seven days after irradiation, the ability of cells to form colonies was assessed. Results are shown as the percentage of plating efficiency and are expressed as mean ± SEM (*n* = 9–12). (**d**) Representative photographs of clone formation for all groups are shown. (**e**) Three days after irradiation, the percentage of cancer cells in particular phases of the cell cycle was analyzed by assessing the DNA content of cancer cells using flow cytometer. Results are shown as mean ± SEM (*n* = 3–4; ^##^
*p* = 0.055; ^###^ *p* = 0.061; ^####^
*p* = 0.076). Data from the co-cultures were compared with data from the respective monocultures using the Student’s *t*-test or the Mann–Whitney *U* test depending on the variable distribution and the homogeneity of variance (* *p* < 0.05, ** *p* < 0.01). Abbreviations: A (A253); A+W (A253 + Wi-38); F (FaDu); F+W (FaDu + Wi-38); W (Wi-38); A+H (A253 + HUVECs); F+H (FaDu + HUVECs); H (HUVECs).

**Figure 2 biomedicines-11-01773-f002:**
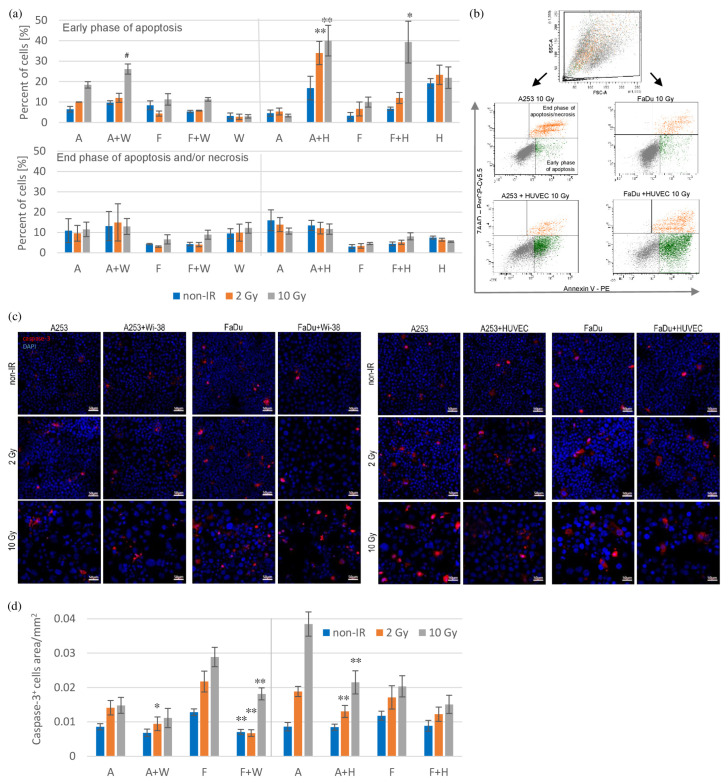
Cell death determination in cell co-cultures compared to cancer cell monocultures after irradiation. (**a**) Three days after irradiation, cells were analyzed using the PE Annexin V Apoptosis Detection Kit I. The percentage of cells in the early and late phases of apoptosis or necrosis was analyzed using flow cytometry analysis. Results are shown as mean ± SEM (*n* = 3–5, ^#^ *p* = 0.058). (**b**) Representative graphs of early and late phases of apoptosis or necrosis are shown from flow cytometry analysis for co-cultures of cancer cells with HUVECs and respective monocultures after 10 Gy irradiation. (**c**) Four days after irradiation, cells were stained with anti-Cleaved Caspase−3 antibody (Texas Red, red) and DAPI (blue). Cells were visualized using confocal microscope. Photographs were taken from 5 randomly chosen fields (20× magn.) per culture in 3 repetitions for each group. Scale bar is 50 µm. Representative photographs for all groups are shown. (**d**) The percentage of the area covered by caspase–3^+^ cells was calculated using ImageJ 1.48v software. Results are shown as mean ± SEM (*n* = 14–15). Data from the co-cultures were compared with data from the respective monocultures by the Student’s *t*-test or the Mann–Whitney *U* test depending on the variable distribution and the homogeneity of variance (* *p* < 0.05, ** *p* < 0.01). Abbreviations: A (A253); A+W (A253 + Wi-38); F (FaDu); F+W (FaDu + Wi-38); W (Wi-38); A+H (A253 + HUVECs); F+H (FaDu + HUVECs); H (HUVECs).

**Figure 3 biomedicines-11-01773-f003:**
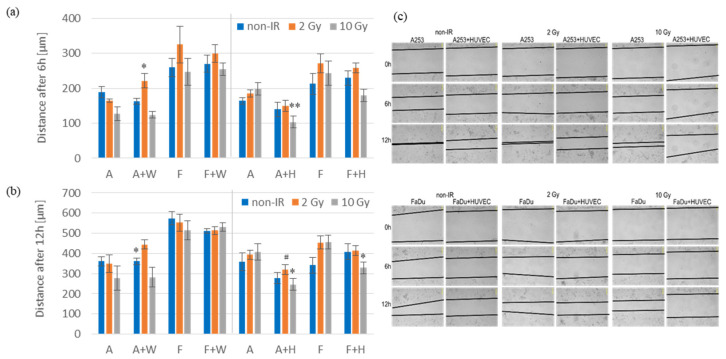
The migration ability of cells in co-cultures compared to monocultures after irradiation. (**a**) One day after irradiation, cell migration was analyzed using the scratch-wound assay. The migration distance was calculated using ZEN 2012 software. Results for the first 6 h of the test are shown as mean ± SEM (*n* = 5–9). (**b**) Results for the first 12 h of the test are shown as mean ± SEM (*n* = 5–9, ^#^ *p* = 0.061). (**c**) Representative photographs for co-cultures of cancer cells with HUVECs and corresponding cancer cell monocultures are shown at 3 time points: 0, 6 and 12 h. Data from the co-cultures were compared with data from the respective monocultures by the Student’s *t*-test (* *p* < 0.05, ** *p* < 0.01). Abbreviations: A (A253); A+W (A253 + Wi-38); F (FaDu); F+W (FaDu + Wi-38); A+H (A253 + HUVECs); F+H (FaDu + HUVECs).

**Figure 4 biomedicines-11-01773-f004:**
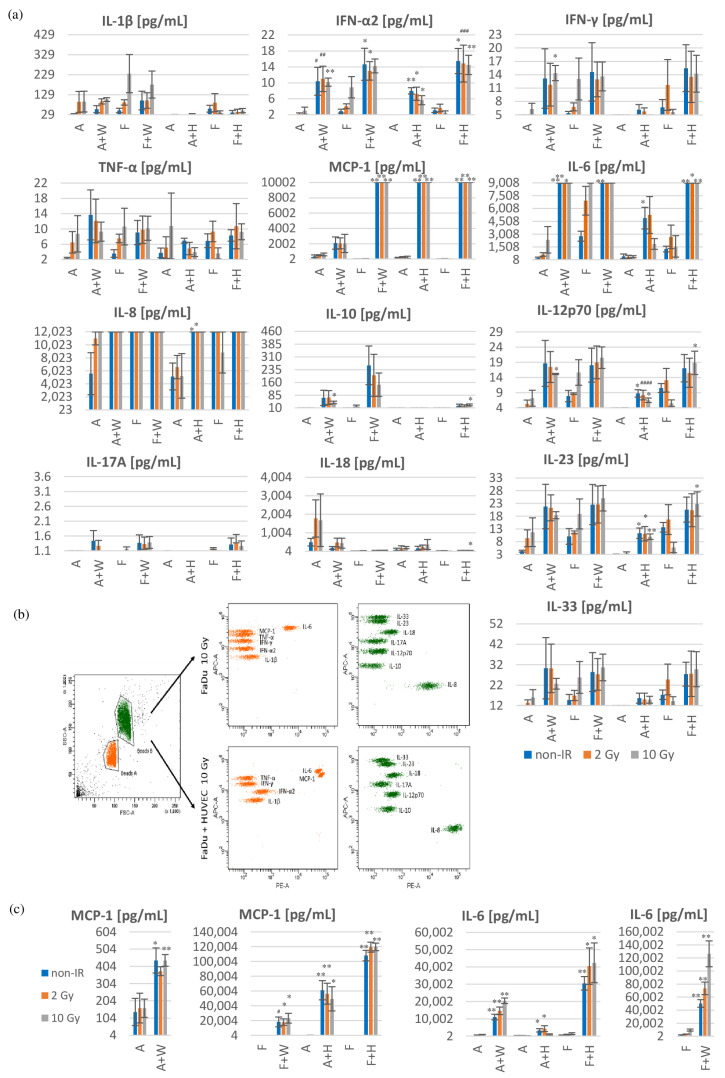
Inflammatory cytokine and chemokine secretion in cell co-cultures compared to cancer cell monocultures after irradiation. (**a**) Inflammatory cytokines and chemokines secreted by the cells into the conditioned medium during 72 h after irradiation were analyzed using the LEGENDplex™ Human Inflammation Panel via flow cytometric analysis. Raw data were analyzed using LEGENDplex^TM^ Data Analysis software. Results are shown as mean ± SEM (*n* = 3–4, ^#^ *p* = 0.074, ^##^ *p* = 0.051, ^###^ *p* = 0.077, ^####^ *p* = 0.055). The minimum values on the y-axis are the lower limit as measured by the LEGENDplex™ Human Inflammation Panel 1 for the individual cytokines and chemokines. The maximum values on the y-axis for IL-6, IL-8, and MCP-1 are the upper limit as measured by the LEGENDplex™ Human Inflammation Panel 1. (**b**) Representative graphs from flow cytometry analysis for co-culture of FaDu with HUVECs and FaDu monoculture after 10 Gy irradiation are shown. (**c**) Additionally, IL-6 and MCP-1 secretion was examined using Human IL-6 ELISA Kit II and ELISA MAXTM Deluxe Set Human MCP-1/CCL2, respectively. Results are shown as mean ± SEM (*n* = 3–4, ^#^ *p* = 0.051). Data from the co-cultures were compared with data from the respective monocultures by the Student’s *t*-test (* *p* < 0.05, ** *p* < 0.01). Abbreviations: A (A253); A+W (A253 + Wi-38); F (FaDu); F+W (FaDu + Wi-38); A+H (A253 + HUVECs); F+H (FaDu + HUVECs).

## Data Availability

All data relevant to the study are included in the article.

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
