# Peer review of "The Effect of Radiotherapy on Cell Survival and Inflammatory Cytokine and Chemokine Secretion in a Co-Culture Model of Head and Neck Squamous Cell Carcinoma and Normal Cells"

_biomedicines, 2023, doi:10.3390/biomedicines11061773_

Round 1
Reviewer 1 Report
This study aimed to assess the effect of radiotherapy on cell survival and inflammatory cytokines and chemokine secretion using HNSCC, fibroblast and endothelial cell cultures.
The major issue with this study is that it lacks novelty. A large portion of the paper refers to prior, well known literature, using classic assays with nothing significantly novel or insightful to add.
The results of this study largely confirm what is already known regarding the subject matter, with many of the observations not being statistically significant. In addition a number of results show no difference with/without radiotherapy which reduces their importance when considering this is a study mainly focusing on the effect of radiotherapy.
The final results regarding IL-6 and MCP-1 remain a hypothesis- it is recommended that the researchers potentially do some validation on this aspect to provide the novelty required from the study.
If the authors reduce the discussion section, focusing less on prior literature and more on what can be deduced from their results, I believe this work could be published in a different journal, one suggestion being 'Anticancer Research'.
The English was good overall, however minor editing could perfect it.
Reviewer 2 Report
Matuszczak et al., contributed the manuscript entitled "The effect of radiotherapy on cell survival and inflammatory 2 cytokines and chemokines secretion in the co-culture model of 3 HNSCC and normal cells". The content is to define the radioresistance of HNSCC cells is due to the co-cultured fibroblasts. The problem of this study is that the novelty is limited because similar studies have been proposed, and the results are only based on in vitro results that also limit the clinical impact. Moreover, this manuscript did not describe the possible solutions to against the induce radioresistance. Other comments are listed below:
1. In figure 1a, the images of gamma-H2AX staining should be present for better comprehension. Does the intensity of gamma-H2AX equal to the amount of gamma-H2AX? Usually the number of gamma-H2AX foci is more important than the intensity of that. Please confirm and show the images.
2. In figure 1c, is the cloning efficiency the plating efficiency? Please use the stansdard term of radiation biology. The resolution of figure 1d is poor, and the survival curves with increased radiation dose is the gold standard to demonstrate the radiosensitivity of cells with different treatments. What test was used for statistical analysis of these data?
3. Were the fibroblasts used in this study CAF? It is not clear how the lung fibroblasts can influence the HNSCC.
4. For wound healing assay, the initial line and final line should be shown in the images. It is not clear about the conclusion from these data as the performance of different cells was also different.
5. In figure 4, did author simply add MCP-1 or IL-6 to demonstrate the effects of these cytokines on inhibition of apotosis and cell migration after HNSCC cells treated with IR?
6. The scale bars are required for cell images.
There are some grammars and writing problems, and have a native English speaker to edit the manuscript.
Round 2
Reviewer 1 Report
With the much improved discussion section and the intention of the authors to further validate and grow this study, I recommend the paper be published.
The quality of the English is acceptable, however it has not been perfected.
Reviewer 2 Report
Authors have addressed my questions.